# Increasing the Efficiency of Food Material Cutting during Inclined and Shear Movements of Knife

**DOI:** 10.3390/ma15010289

**Published:** 2021-12-31

**Authors:** Oleg V. Ageev, Andrzej Dowgiałło, Monika Sterczyńska, Joanna Piepiórka-Stepuk, Natalia V. Samojlova, Marek Jakubowski

**Affiliations:** 1Department of Food and Refrigeration Machines, Mechanics and Technology Faculty, Kaliningrad State Technical University, 236022 Kaliningrad, Russia; oleg.ageev@klgtu.ru; 2Department of Processing Technology and Mechanization, National Marine Fisheries Research Institute, 81-332 Gdynia, Poland; adowgiallo@mir.gdynia.pl; 3Department of Food Industry Processes and Facilities, Faculty of Mechanical Engineering, Koszalin University of Technology, 15-17 Raclawicka Str., 75-620 Koszalin, Poland; monika.sterczynska@tu.koszalin.pl (M.S.); joanna.piepiorka@tu.koszalin.pl (J.P.-S.); 4Department of Processing Equipment Engineering, Mechanics and Technology Faculty, Kaliningrad State Technical University, 236022 Kaliningrad, Russia; vertige_@mail.ru

**Keywords:** cutting, strength, resistance, shape, knife, edge, sharpness, rheology, viscoelasticity

## Abstract

Mathematical models for predicting the resistance forces that are developed during the inclined and sliding cutting of food materials have been developed. The dependence of the actual cutting angle on the angle of inclination and sliding speed of the cutting edge at various sharpening angles have been investigated. For the inclined cutting mode, the dependence of the useful resistance force on the cutting speed has been determined at various angles of inclination of the cutting edge and designed sharpening angles. For the sliding cutting mode, the dependence of the useful resistance force on the feeding speed has been demonstrated at various sliding speed values and designed knife sharpening angles. The dependence of the transformed dimensionless sharpness of the knife on the angle of inclination of the cutting edge and the sliding speed has been established for different constructional sharpness values of the knife. The results of the study indicate that the useful resistance force is significantly reduced during the inclined and sliding cutting processes when compared with the normal cutting process, and a change in the pattern of fiber destruction, which significantly increases the cutting efficiency of cutting tools, is obtained.

## 1. Introduction

There has been an increasing demand to explore measures that can increase the efficiency and speed of the technological processes involved in the cutting of raw materials. Modern food production techniques should work toward achieving this goal by developing highly efficient cutting tools characterized by the highest cutting efficiency. The advantages of such processes include a reduction in the energy costs for processing, improved quality of the finished product, an increase in productivity, a reduction in metal consumption, the choice of different rational cutting modes, the use of wear-resistant knives with antifriction coatings, and so on. The efficiency of the cutting process can be enhanced by reducing the resistance forces that develop when the material is cut into small portions. This can be achieved by the geometric and kinematic transformation of the cutting force components, particularly the knife sharpening angle and the sharpness of the cutting edge of the blade.

Dowgiałło (2005) proposed fundamental models for calculating the resistance forces that arise during the cutting of fish [1]. Other authors investigated and optimized various designs to increase the efficiency of high-speed cutting tools used for food products [2]. Atkins (2006) proposed theoretical concepts and determined the optimal parameters that influence the cutting process of the cutting equipment [3]. Yet, in other research developed empirical models of the cutting process based on the influence of frictional forces and the asymmetry profile of the cutting body [4]. The works of other authors focused on the processes involved in the cutting and grinding of meat in tops and analyzed the effects of the sharpening angles of the working bodies on the cutting forces [5]. They further developed a method to measure the sharpness of the blade. Researchers studied the influence of the shape of the cutting edge and the sharpness of knives on the cutting forces in beef [6]. Brown et al. (2005) investigated different tools for cutting cheese, bacon, and beef and studied the influence of temperature on the cutting process by performing experiments at various temperatures [7]. Other authors constructed empirical models to illustrate the mechanisms underlying the cutting process of food products [8]. Van Vliet (2014) experimentally investigated the deformation and fracture processes observed during the cutting of viscous products [9].

Spagnoli et al. (2019) compared the effects of blade inclination and friction on hard and soft polymeric materials [10]. Scientists, too, also performed a comparative study to evaluate the influence of biochemical composition and functional and rheological properties of fresh meat obtained from fish, squid, and shrimp on cutting techniques [11]. Pagani and Perego (2015) used solid-shell finite element models to illustrate explicit dynamics simulation of blade cutting of thin elastoplastic shells using directional cohesive elements [12]. Goh et al. (2005) analyzed the mechanics involved in the wire cutting of cheese [13]. Other authors studied the effects of critical stress and critical distance on crack propagation in the cutting models of cheese [14].

Holl et al. (2018) examined the influence of superimposed vibrations on reducing friction in food samples [15]. Xiao-Ping et al. (2019) analyzed the mechanical and fracturing behavior in polymethyl methacrylate (PMMA) specimens containing multiple three-dimensional (3D) embedded flaws under uniaxial compression [16]. Ogunsina and Bamgboye (2013) measured fracture resistance in cashew nuts that were subjected to preshelling treatment [17]. Sadowska et al. (2013) examined the effect of seed size and microstructure on the mechanical properties and frictional behavior of nuts [18]. Kasperowicz et al. (2019) determined the effect of the supply pressure of water on the cutting process of fish using a high-pressure water stream by taking into account the cutting place and diameter of the water nozzle [19]. Sridhar and Sommer (2013) adopted finite element method (FEM) simulation methodology to analyze the fracture mechanism in fibrous food [20]. Yildiz et al. (2019) described ultrasonic cutting as a new technology that can produce fresh-cut red delicious and golden delicious apples with improved quality [21].

Nawaz et al. (2019) evaluated the physicochemical, textural, and sensory qualities of red fish meat-based fried snacks. Usydus Z. and Szlinder-Richert J. presented a review of the functional properties of fish and fish products [22]. Wilson et al. (2020) considered the use of customized shapes for chicken meat-based products and conducted feasibility studies on 3D-printed nuggets [23]. Murthy et al. (2017) determined the effect of sucrose and sorbitol on the rheological properties of washed and unwashed tilapia (*Oreochromis mossambicus*) fish meat [24]. Nelson et al. (2020) proposed a mathematical model for the meat cooking process [25]. Other authors evaluated the effect of ultrasound treatment on the functional properties of reduced salt chicken breast meat batter [26]. Malakizadi et al. (2017) used friction models to study the influence of friction on FE simulation results of the orthogonal cutting process [27]. Ageev et al. (2021) proposed a set of theoretical models for calculating contact pressures and resistance forces during the cutting of fish [28].

The results by our previous papers show that the reduction of undesirable energy consumption for cutting by changing the geometry of the knife was achieved. This was done by reducing the sharpening angle and the thickness of the knife, as well as by constructively introducing rear inclined edges and eliminating the side edges. The reduction of friction forces was facilitated by a decrease in the roughness of the edges, which was achieved by polishing and the use of anti-friction coatings [29].

Equipping the knife with back edges lead to the appearance of a co-directional reactive force, which reduced the resulting resistance force, as well as the appearance of an additional deformational friction force applied along the length of the rough edge. However, as simulations show, the beneficial effect of reducing the resulting resistance force significantly exceeded the additional friction losses [30].

The solutions to optimization problems show that it is possible to determine the design angles of sharpening the edges of the knife such that the resistance force and deformation forces of friction will be minimal [31]. In the case of using a knife without edges, which is a thin cutting string, its efficiency is practically equal to unity, which means the consumption of the supplied energy in the useful destruction of the material without harmful losses. Along with an increase in the efficiency of the knife’s action due to the introduction of rear inclined edges and the elimination of side edges, a significant direction in reducing harmful losses is reducing the thickness and angle of sharpening of the knife, as well as improving the sharpness of the cutting edge. This approach leads to a significant reduction in form resistance forces [32].

The paper [32] also provides an approach for determining the specific force for cutting an elementary knife. It has been found that each elementary knife has a coefficient of efficiency, which depends on the actual cutting angle and the depth of immersion of the elementary knife into the material being cut. The immersion depth of an elementary knife determines the line of its force interaction with the material and, accordingly, the magnitude of the forces of harmful and useful resistances. The specified line is calculated as the distance from the cutting edge of the knife along the line of its net speed to the surface of the material. Since the resulting speeds of different points of the circular knife differ, the immersion depth of an elementary knife is defined as the length from the edge of the knife to the surface of the material along a curve, the tangents to which, at any point, coincide with the vectors of the knife’s velocity relative to the material at the same point.

Although a considerable amount of research has been carried out with regard to the efficacy of different cutting techniques, currently there is a lack of mathematical models that enable the determination of the resistance forces during inclined and sliding cutting with a knife. Furthermore, it is necessary to design technically and economically feasible food processing equipment with efficient cutting tools. Therefore, the task of performing a theoretical analysis to improve the efficiency of the cutting process of food materials is highly relevant in this context.

The objectives of the present study are to perform a detailed analysis of the resistance forces that are developed during the inclined and sliding movement of the knife and to elaborate on the methods used to increase the efficiency of the cutting process of food products.

## 2. Theoretical Modeling of Resistance Force during the Geometric Transformation of a Knife Sharpening Angle

As shown in a previous work by Ageev et al. (2021) [29], the resistance force developed during the normal cutting of food material significantly depends on the thickness and sharpening angle of the knife (1):(1)$$F_{1}=\frac{\left(1-\exp\left(k\cdot \delta /tg\alpha \right)\right)\cdot \xi^{2}\cdot \eta \cdot v\cdot l_{k}}{k\cdot E_{1}^{2}\cdot l}\cdot tg^{2}\alpha +\frac{\delta \cdot \xi^{2}\cdot \eta \cdot v\cdot l_{k}}{E_{1}^{2}\cdot l}\cdot tg\alpha +\frac{\delta^{2}\cdot \xi \cdot l_{k}}{2\cdot l}$$

A reduction in the thickness of the blade leads to a significant decrease in the specified force. The resistance force also decreases with a decrease in the sharpening angle, but to a lesser extent. Therefore, in order to reduce the energy costs associated with harmful resistance forces, it is advisable to reduce the constructional thickness and sharpening angle of the cutting tool. However, this is not always feasible because it results in a noticeable deterioration in the strength of the knife. In addition, a decrease in the sharpening angle of the knife, while maintaining its thickness, leads to an increase in the height of the front inclined edges, that is, $h_{m}=\delta /tg\alpha$. In normal (chopping) cutting, the angle of inclination of the cutting edge to the direction of movement of the knife is a right angle, and, therefore, the actual cutting angle coincides with the constructional sharpening angle, that is, $\alpha_{\phi }=\alpha$. The situation changes drastically if the cutting edge is inclined toward the direction of knife movement. In this case, a geometric transformation of the actual cutting angle occurs, which allows achieving the desired effect of reducing the sharpening angle of the knife. This mode of inclined cutting is observed when a guillotine knife with an inclined edge is used.

Let us consider the diagram illustrating the cutting of a material with a knife having an inclined cutting edge and one-sided sharpening (Figure 1a). From the diagram, the relationship between the constructional sharpening angle (α) and the actual cutting angle ($\alpha_{\phi }$) can be determined by considering the triangles ABC and $AB_{1}C_{1}$. From the ratios we get the dependence Equation (2):(2)$$tg\alpha =\frac{B_{1}C_{1}}{AC_{1}},\;tg\alpha_{\phi }=\frac{BC}{AC}=\frac{B_{1}C_{1}}{AC},\;∠AC_{1}C=90{}^{\circ},\;\mathrm{and}\;\;AC_{1}=AC\cdot \cos\gamma_{H}tg\alpha_{\phi }=tg\alpha \cdot \cos\gamma_{H}$$

Figure 2a shows the dependence of the actual cutting angle on the angle of inclination of the cutting edge at different constructional sharpening angles of the knife, while Figure 2b shows the dependence of the specified angle on the sliding speed at various constructional sharpening angles of the knife in the sliding cutting mode, which is discussed in the following section.

The mathematical model (Equation (1)) for the inclined cutting mode is given as follows:(3)$$F_{1}^{t}=\frac{\xi \cdot l_{k}}{l}\cdot \left(\frac{\left(1-\exp\left[\frac{k\cdot \delta }{tg\alpha \cdot \cos\gamma_{H}}\right]\right)\cdot \xi \cdot \eta \cdot v\cdot tg^{2}\alpha \cdot \cos^{2}\gamma_{H}}{k\cdot E_{1}^{2}}+\frac{\delta \cdot \xi \cdot \eta \cdot v\cdot tg\alpha \cdot \cos\gamma_{H}}{E_{1}^{2}}+\frac{\delta^{2}}{2}\right)$$

In special cases when $\gamma_{H}=0$, Equation (3) coincides with Equation (1): $F_{1}^{t}=F_{1}$. Figure 3 shows the dependence of the resistance force of the knife shape on the cutting speed at different angles of inclination of the cutting edge and constructional angles of sharpening.

## 3. Theoretical Modeling of Resistance Force during the Kinematic Transformation of the Knife Sharpening Angle and the Sharpness of the Cutting Edge of the Knife

A more drastic transformation of the sharpening angle occurs during the sliding cutting of material when, in addition to the working movement, an additional force is communicated to the cutting tool. This situation is observed due to the simultaneous movement of the knife in two directions: normal to the cutting edge at speed $v_{p}$ and parallel to the edge at speed $v_{o}$. The cutting of fish with a continuously moving knife is mechanically carried out by mounting an endless belt knife on pulleys. Thus, an elementary knife penetrates the material with the resulting speed v at an angle $\gamma_{c}$ normal to the blade surface (Figure 1b). In this case, the angle $\gamma_{c}$ is the sliding angle of the knife, and the ratio $K_{\gamma }=v_{o}/v_{p}=tg\gamma_{c}$ is the sliding coefficient. The resulting speed of an elementary knife will be $v=\sqrt{v_{p}^{2}+v_{o}^{2}}$. From the triangles $AA_{2}B_{1}$ and $AA_{1}B$, we derive the expression for the actual cutting angle that is formed during the sliding cutting process: $tg\alpha_{\phi }=tg\alpha \cdot \cos\gamma_{c}=tg\alpha \cdot v_{p}/\sqrt{v_{p}^{2}+v_{o}^{2}}$.

Where $\gamma_{c}$ is the sliding angle of the cutting edge and also the angle of the kinematic ascent of the plane of the actual elementary knife relative to the plane of the constructional elementary knife.

The mathematical model (Equation (1)) for sliding-cutting mode is transformed into the following expression:(4)$$F_{1}^{k}=\frac{\xi \cdot l_{k}}{l}\cdot \{\overset{}{}-\left(\overset{}{}1-\exp\left[-\frac{\left(E_{0}+E_{1}\right)\cdot \delta \cdot \sqrt{v_{p}^{2}+v_{o}^{2}}}{\eta \cdot v_{p}^{2}\cdot tg\alpha }\right]\overset{}{}\right)\cdot \frac{\xi \cdot \eta^{2}\cdot v_{p}^{4}\cdot tg^{2}\alpha }{E_{1}^{2}\cdot \left(E_{0}+E_{1}\right)\cdot \left(v_{p}^{2}+v_{o}^{2}\right)}+\;+\frac{\delta \cdot \xi \cdot \eta \cdot v_{p}^{2}\cdot tg\alpha }{E_{1}^{2}\cdot \sqrt{v_{p}^{2}+v_{o}^{2}}}+\frac{\delta^{2}}{2}\overset{}{}\}$$

In special cases when $v_{o}=0$, $v=v_{p}$, and $\gamma_{c}=0$, Equation (5) is the same as Equation (1): $F_{1}^{k}=F_{1}$. When $v_{p}\to 0$ or $v_{o}\to \infty$, $F_{1}^{k}\to \xi \cdot l_{k}\cdot \delta^{2}/\left(2\cdot l\right)$. Figure 4 shows the dependence of the shape resistance force on the feed speed at various values of the sliding speed and constructional sharpening angles of the knife.

In inclined and sliding cutting processes, along with the transformation of the sharpening angle of an elementary knife, a transformation of the profile of the cutting edge is also observed, which contributes to an improvement in the actual sharpness of the blade. The cross-section of the cutting edge is a circular arc with a radius ρ. During inclined and sliding cutting, the plane of the actual cutting angle of the elementary knife rises relative to the plane of the constructional angle of sharpening, and a section of its edge is transformed and takes the form of a longitudinal segment of an elliptical cylinder. The arc of the sharp vertex of the ellipse in the case of inclined cutting has a radius $\rho_{H}=\rho \cdot \cos\gamma_{H}$ and, in the case of sliding cutting, $\rho_{c}=\rho \cdot \cos\gamma_{c}=\rho \cdot v_{p}/\sqrt{v_{p}^{2}+v_{o}^{2}}$.

Let us estimate the impact of the transformation of the knife sharpness on the force of useful resistance. Figure 5a shows the dependence of the transformed dimensionless knife sharpness on the angle of inclination of the cutting edge at various dimensionless constructional sharpness values of the blade. Figure 5b shows the dependence of the specified sharpness on the dimensionless sliding speed at various dimensionless constructional sharpness values of the knife.

According to a previous work by Ageev et al. (2021) [29] we write the expression for the dimensionless useful resistance force developed due to the longitudinal stretching of the fibers during normal cutting of the material with a knife whose cutting edge has not been transformed (5):(5)$${\bar{T}}_{3}=\frac{2\cdot \bar{\rho }\cdot \cos^{2}\alpha }{\left({\bar{l}}_{a}+\bar{\rho }\cdot \alpha_{r}\right)}\cdot \{\overset{}{}\frac{3}{5\cdot {\bar{l}}_{a}}\cdot \left(\overset{}{}{\bar{h}}_{u}^{2}+2\cdot \bar{v}\cdot e_{01}\cdot {\bar{h}}_{u}-2\cdot {\bar{v}}^{2}\cdot e_{01}\overset{}{}\right)+\;+\exp\left(-\frac{1}{\bar{v}}\right)\cdot (\bar{\rho }\cdot \frac{\bar{\rho }\cdot \alpha_{r}^{2}-2\cdot \bar{\rho }\cdot \alpha_{r}\cdot \cos\alpha -{\bar{l}}_{a}\cdot \cos\alpha }{\left({\bar{l}}_{a}+\bar{\rho }\cdot \cos\alpha \right)}+\frac{6\cdot {\bar{v}}^{2}\cdot e_{01}}{5\cdot {\bar{l}}_{a}}+\bar{\rho }\cdot \alpha_{r})\overset{}{}\}$$

In case of inclined cutting, considering ${\bar{\rho }}_{H}=\bar{\rho }\cdot \cos\gamma_{H}$, $\alpha_{\phi }=arctg\left(tg\alpha \cdot \cos\gamma_{H}\right)$, and $\alpha_{rt}=\frac{\pi \cdot \left(90{}^{\circ}-\alpha_{\phi }\right)}{180{}^{\circ}}=\frac{\pi \cdot \left(90{}^{\circ}-arctg\left(tg\alpha \cdot \cos\gamma_{H}\right)\right)}{180{}^{\circ}}$,

Equation (6) is converted to a simplified form, without taking the effect of spatial stretching of the fibers into account:(6)$${\bar{T}}_{3}^{t}=\frac{2\cdot {\bar{\rho }}_{H}\cdot \cos^{2}\alpha_{\phi }}{\left({\bar{l}}_{a}+{\bar{\rho }}_{H}\cdot \alpha_{rt}\right)}\cdot \{\overset{}{}\frac{3}{5\cdot {\bar{l}}_{a}}\cdot \left(\overset{}{}{\bar{h}}_{u}^{2}+2\cdot \bar{v}\cdot e_{01}\cdot {\bar{h}}_{u}-2\cdot {\bar{v}}^{2}\cdot e_{01}\overset{}{}\right)+\;+\exp\left(-\frac{1}{\bar{v}}\right)\cdot ({\bar{\rho }}_{H}\cdot \frac{{\bar{\rho }}_{H}\cdot \alpha_{rt}^{2}-2\cdot {\bar{\rho }}_{H}\cdot \alpha_{rt}\cdot \cos\alpha_{\phi }-{\bar{l}}_{a}\cdot \cos\alpha_{\phi }}{\left({\bar{l}}_{a}+{\bar{\rho }}_{H}\cdot \cos\alpha_{\phi }\right)}+\;+\frac{6\cdot {\bar{v}}^{2}\cdot e_{01}}{5\cdot {\bar{l}}_{a}}+{\bar{\rho }}_{H}\cdot \alpha_{rt})\overset{}{}\}$$

With sliding cutting, taking into account ${\bar{\rho }}_{c}=\bar{\rho }\cdot \cos\gamma_{c}=\bar{\rho }\cdot {\bar{v}}_{p}/\sqrt{{\bar{v}}_{p}^{2}+{\bar{v}}_{o}^{2}}$, $\alpha_{\phi }=arctg\left(tg\alpha \cdot \cos\gamma_{c}\right)=arctg\left(tg\alpha \cdot {\bar{v}}_{p}/\sqrt{{\bar{v}}_{p}^{2}+{\bar{v}}_{o}^{2}}\right)$, and $\alpha_{rk}=\left[\pi \cdot \left(90{}^{\circ}-\alpha_{\phi }\right)\right]/180{}^{\circ}$, Equation (6) can also be written as follows (7):(7)$${\bar{T}}_{3}^{k}=\frac{2\cdot {\bar{\rho }}_{c}\cdot \cos^{2}\alpha_{\phi }}{\left({\bar{l}}_{a}+{\bar{\rho }}_{c}\cdot \alpha_{rk}\right)}\cdot \{\overset{}{}\frac{3}{5\cdot {\bar{l}}_{a}}\cdot \left(\overset{}{}{\bar{h}}_{u}^{2}+2\cdot {\bar{v}}_{p}\cdot e_{01}\cdot {\bar{h}}_{u}-2\cdot {\bar{v}}_{p}^{2}\cdot e_{01}\overset{}{}\right)+\;+\exp\left(-\frac{1}{{\bar{v}}_{p}}\right)\cdot ({\bar{\rho }}_{c}\cdot \frac{{\bar{\rho }}_{c}\cdot \alpha_{rk}^{2}-2\cdot {\bar{\rho }}_{c}\cdot \alpha_{rk}\cdot \cos\alpha_{\phi }-{\bar{l}}_{a}\cdot \cos\alpha_{\phi }}{\left({\bar{l}}_{a}+{\bar{\rho }}_{c}\cdot \cos\alpha_{\phi }\right)}+\;+\frac{6\cdot {\bar{v}}_{p}^{2}\cdot e_{01}}{5\cdot {\bar{l}}_{a}}+{\bar{\rho }}_{c}\cdot \alpha_{rk})\overset{}{}\}$$

Figure 6 shows the dependence of the dimensionless useful resistance force on the dimensionless cutting speed at different angles of inclination of the cutting edge and constructional angles of sharpening. Figure 7 shows the dependence of the specified force on the dimensionless constructional sharpness of the knife at various angles of inclination of the cutting edge and constructional angles of sharpening. Figure 8 shows the dependence of the dimensionless useful resistance force on the dimensionless feed speed of the material at various dimensionless sliding speed values and constructional angles of sharpening. Figure 9 shows the dependence of the specified force on the dimensionless constructional sharpness of the knife at various dimensionless sliding speed values and constructional angles of sharpening. The contour plots (Figure 10) show the dependence of the specified force on the dimensionless constructional sharpness of the knife, the angle of inclination, and the sliding speed of the cutting edge.

## 4. Discussion

Figure 2a shows that the actual cutting angle decreases nonlinearly if the angle of inclination of the cutting edge increases and can take values within the limits $0<\alpha_{\phi }\leq \alpha$. As a special case, when $\alpha_{\phi }=\alpha$, the actual cutting angle attains a maximum value at the maximum value of the angle of inclination, $\gamma_{H}=0$, which is observed with normal cutting. When the constructional knife sharpening angle is 10° and the angles of inclination of the cutting edge are 10°, 20°, 30°, and 40°, the values of the actual cutting angle are 9.854°, 9.074°, 8.206°, and 7.844°, respectively. When the constructional knife sharpening angle is 40° and the angles of inclination of the cutting edge are 10°, 20°, 30°, and 40°, the values of the actual cutting angle are 39.754°, 37.253°, 36.236°, and 33.847°, respectively.

A significant reduction in the actual cutting angle is noticed with larger values of the angle of inclination of the cutting edge. However, previous results have shown that an increase in the angle of inclination results in the development of a force that shifts the material layer relative to the knife. This leads to a certain decrease in the cutting accuracy and the curvature of the blade trajectory in the material, which requires additional methods to cut the sample perfectly. In these conditions, it is advisable to use a knife with a wavy shape of its cutting edge, so that the actual cutting angle can be further reduced without significantly increasing the angle of inclination of the tool toward the direction of its movement. The greater the angle of inclination of the curly teeth of the wavy edge, the lower the value of the actual cutting angle, which, in this case, will depend on the point where the edge interacts with the material. At points near the top of the wave, the angle of inclination ($\gamma_{H}$) increases, whereas at the base of the wave, the angle will be small. Even when the wavy edge penetrates into the material without tilting the knife, the average actual cutting angle will be significantly less than the constructional sharpening angle.

Figure 2b shows that the actual cutting angle decreases nonlinearly with increasing sliding speed and can take values within the limits $0<\alpha_{\phi }\leq \alpha$. In special cases, when $\alpha_{\phi }=\alpha$, the actual cutting angle achieves maximum value at zero sliding speed, $v_{o}=0$, which is observed with the normal cutting process. At zero feed speed ($v_{p}$), the actual cutting angle is zero and the sliding angle is 90°; the cutting of the material is not possible and the knife slides along the layer without being immersed in it. The same phenomenon is observed with an unlimited increase in the sliding speed of the knife, $v_{o}\to \infty$. When the feed speed is 0.2 m·s^−1^, the constructional knife sharpening angle is 10°, the sliding speeds of the cutting edge are 0.2 m·s^−1^, 0.4 m·s^−1^, 0.6 m·s^−1^, and 0.8 m·s^−1^, the values of the actual cutting angle are found to be 7.054°, 4.326°, 2.594°, and 2.147°, respectively. When the feed speed is 0.2 m·s^−1^, the constructional knife sharpening angle is 40°, the sliding speeds of the cutting edge are 0.2 m·s^−1^, 0.4 m·s^−1^, 0.6 m·s^−1^, and 0.8 m·s^−1^, the values of the actual cutting angle are found to be 30.054°, 20.126°, 15.104°, and 12.038°, respectively.

Figure 3 shows that, with an increase in the inclination of the cutting edge to the surface of the material, the resistance force of the knife shape decreases at the same cutting speed. A reduction in the constructional sharpening angle also leads to a decrease in the specified force.

Therefore, the use of a guillotine knife with an inclined edge and the transition to inclined cutting, while maintaining the strength characteristics of the knife, reduces the effects of harmful resistance forces, increase the efficiency of the cutting tool and reducing the energy costs associated with the cutting process. Inclined cutting has certain advantages over normal cutting, which can be attributed to a decrease in the counter components producing harmful resistance forces, the sawing effect of the micro teeth of the edge on the product, less crushing of the material and hence loss of nutrients, and better preservation of taste and marketability.

Figure 4 illustrates that, with an increase in the sliding speed of the cutting edge relative to the surface of the material, the shape resistance force also decreases significantly. Generally, in complex cutting processes, the transformation of the sharpening angle of an elementary knife will depend on the ratio of speeds, the constructional angle of sharpening, the inclination of the cutting edge, and the direction of relative motion.

Figure 5a shows that, with an increase in the angle of inclination of the cutting edge, the dimensionless sharpness of the knife decreases significantly nonlinearly. Figure 5b also illustrates that, with an increase in the sliding speed of the cutting edge during material destruction, the actual sharpness of the blade significantly decreases and tends to attain a value of zero with an unlimited increase in the specified speed. Thus, at the specified cutting conditions, significant sharpening of the blade takes place due to the transformation of its edge.

Figure 6 demonstrates that, with an increase in the angle of inclination of the cutting edge, the dimensionless useful resistance force decreases significantly nonlinearly and tends to attain a value of zero at $\gamma_{H}\to 90{}^{\circ}$. With an increase in the sharpening angle of the knife, the specified force is markedly reduced, as a result of a decrease in the contact area of the knife edge with the material fibers. Figure 8 illustrates that with an increase in the dimensionless sliding speed of the cutting edge, the dimensionless useful resistance force also decreases nonlinearly and tends to become zero with an unlimited increase in the indicated speed, $v_{o}\to \infty$.

Figure 7 and Figure 9 show that an increase in the constructional sharpness of the blade leads to a significant increase in the specified force; however, inclined cutting and kinematic transformation of the cutting edge reduce the actual sharpness and thereby reduce the dimensionless force.

The simulation results illustrated in Figure 10 demonstrate that, with inclined cutting and kinematic transformation of the cutting edge, the useful resistance force is reduced much more significantly than the shape resistance force under similar conditions. This finding indicates that in inclined and sliding cutting processes, the reduction in the energy costs is mainly due to a decrease in the useful resistance forces acting on the cutting edge.

A significant increase in the efficiency of the knife and a decrease in energy consumption is observed with the transformation of the sharpening angle and sharpness of the knife. This finding can be exploited to develop an appropriate working tool in fish processing equipment, for example, by installing a high-speed tape knife at an angle to the direction of movement of the raw materials. However, using such knives for the processing of fish requires the calculation of the forces acting on the thin working blade of the cutting tool when it is installed on the drive.

It should be noted that during the transition from normal cutting mode to the inclined and sliding cutting mode, the efficiency of the knife increases not only due to a decrease in the harmful resistance forces and an improvement in the sharpness of the cutting edge, but also due to a significant change in the pattern of destruction of fibers. In particular, in sliding cutting, the destruction of the muscle fibers of the material occurs at a much faster rate at the same depth of immersion of the knife, since deformation due to the longitudinal extension of the filaments increases significantly and their rupture occurs in a shorter time interval of the blade movement.

In addition, minute notches present on the cutting edge of the blade and cuts, resulting from the sharpening of the knife with abrasive tools, contribute to a reduction in the resistance forces during inclined and sliding cutting. At the time of sharpening, the individual grains of the abrasive wheel form the smallest grooves along the trajectory of these grains. The grooves converging at the blade form small notches, divorced from one and other and by a certain step. The notches, which act like a saw during the cutting process, help in the removal of particles from the material and facilitate the immersion of the edge into the muscle tissue.

The simulation results are in good agreement with several studies by Vliet (2014) [9], Goh et al. (2005) [13,14]. For a more accurate calculation of cutting forces, one is advised to model the friction force between blade and material. Friction is indicated by force profiles and shear behavior. In addition, the effect of deformation of the portion should be taken into account, which is confirmed by the results of experiments performed by Spagnoli et al. (2019) [10]. For the accurate calculation of shear stress friction according to Atkins et al. (2006) [3] and Spagnoli et al. (2019) [10], it is necessary to quantify the contact surface between the blade and the food and determine how it is influenced by the geometric parameters of the knife inclination, and to determine the sliding speed of the cutting. Presumably, the decrease in blade sliding force is the result of the relative vertical movement of knife compared to the horizontal feed rate (Figure 1b), which is in good agreement with the data by Brown et al. (2005) [7].

## 5. Conclusions

This study has theoretically confirmed that the efficiency of the fish cutting process can be improved by adopting both inclined and sliding cutting modes, emphasizing the reduction of harmful and useful resistance forces. Losses due to viscoelastic resistance and friction are minimized due to the transformation of the sharpening angle and sharpness of the blade, partial transfer of pressure and frictional forces from the normal direction to the tangential one, sawing action of the edge on the muscle fibers, and a faster onset of material destruction. Since a decrease in the constructional sharpening angle and an improvement in the sharpness of the cutting edge depend on the mechanical characteristics of the knife, the indicated effects of transformation of the knife geometry on the cutting process cannot be established in all situations.

Numerical computation has shown that an increase of the angle of inclination of the cutting edge leads to a non-linear decrease in the actual cutting angle. In addition, increases in the cutting speed of the knife have resulted in an asymptotic reduction in the actual cutting angle. Due to the fact that a decrease in the actual cutting angle and constructional sharpness entails a decrease in the cutting resistance forces, the undesirable resistance forces decrease with an increase in the angle of inclination of the cutting edge and an increase in the sliding speed. This allows reducing energy costs when cutting food materials.

The above results show that it would be tempting to simultaneously use the effects of geometric and kinematic knife transformations. This can be achieved by using a circular cutter. Such a circular knife combines the positive effects of geometric and kinematic transformations of the shape of its profile. Their use in food machines, along with oblique lamellar and band knives, is expedient due to the simplicity of realizing a uniform rotational motion with the required speed. When cutting with a circular knife, a passing or counter movement of its edge in relation to the direction of material feeding is possible. When analyzing the parameters of such a tool, it is necessary to consider not only its static geometry, but also its kinematic one. During the kinematic transformation of the disc cutter chamfer during its rotation and immersion in the material, the shape of the profile of the elementary knife changes, as does the actual cutting angle. In this case, the rectilinear front inclined edge acquires a curved shape.

## Figures and Tables

**Figure 1 materials-15-00289-f001:**
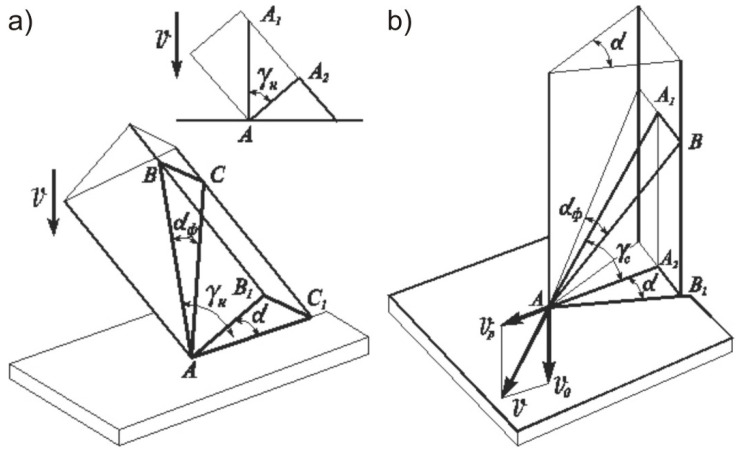
Diagrams of inclined (**a**) and sliding (**b**) cutting of material with a linear knife (v—resulting cutting speed; $v_{p}$—horizontal feed speed; $v_{o}$—vertical sliding speed; α—constructive sharpening angle; $\alpha_{\phi }$—actual sharpening angle, same as kinematic effective cutting angle; $\gamma_{H}$—the angle of inclination of the cutting edge to the direction of movement of the knife; $\gamma_{c}$—the sliding angle of the cutting edge, and also the angle of the kinematic ascent of the plane of the actual elementary knife relative to the plane of the constructional elementary knife; A—point of contact between the edge of the knife and the surface of the material).

**Figure 2 materials-15-00289-f002:**
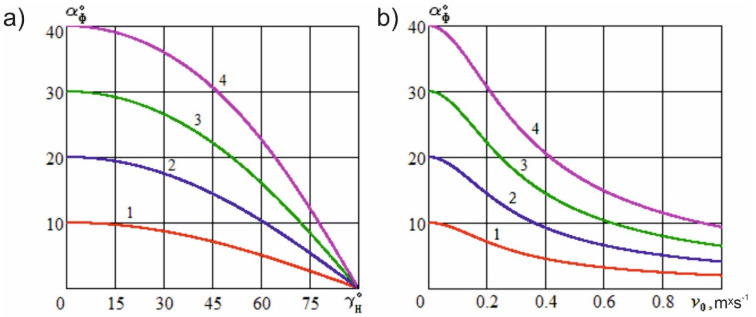
Dependence of the actual cutting angle on the angle of inclination of the cutting edge (**a**) and sliding speed (**b**) at various constructional knife sharpening angles ($v_{p}=0.2$ m·s^−1^): 1—α=10°; 2—α=20°; 3—α=30°; 4—α=40°.

**Figure 3 materials-15-00289-f003:**
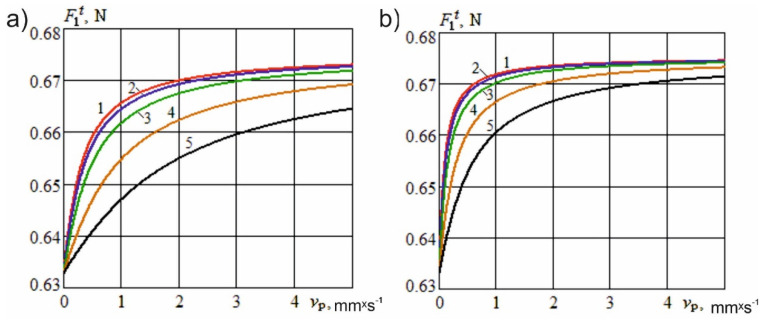
Dependence of the resistance force of the knife shape on the cutting speed at different angles of inclination of the cutting edge. $E_{0}=1.5\times 10^{5}$
N×m^−2^; $e_{01}=15$; δ=3 mm: (**a**)—α=10°; (**b**)—α=30°; 1—$\gamma_{H}=0$; 2—$\gamma_{H}=30{}^{\circ}$; 3—$\gamma_{H}=50{}^{\circ}$; 4—$\gamma_{H}=70{}^{\circ}$; 5—$\gamma_{H}=80{}^{\circ}$.

**Figure 4 materials-15-00289-f004:**
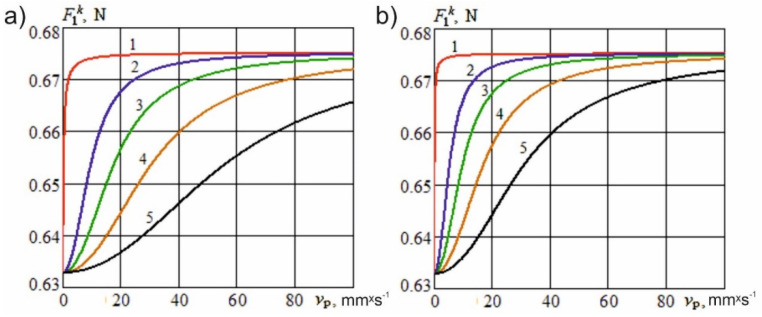
Dependence of the resistance force of the knife shape on the feed speed at various values of the sliding speed ($E_{0}=1.5\times 10^{5}$
N×m^−2^; $e_{01}=15$; δ=3 mm): (**a**)—α=10°; (**b**)—α=30°; 1—$v_{o}=0$; 2—$v_{o}=0.3$ m·s^−1^; 3—$v_{o}=1$ m/s; 4—$v_{o}=3$ m/s; 5—$v_{o}=10$ m/s.

**Figure 5 materials-15-00289-f005:**
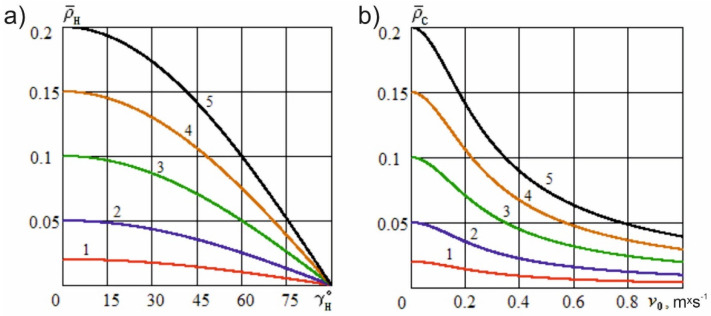
Dependence of the transformed dimensionless knife sharpness on the angle of inclination of the cutting edge (**a**) and sliding speed (**b**) for different values of the dimensionless constructional sharpness of the blade (m/s): 1—$\bar{\rho }=0.02$; 2—$\bar{\rho }=0.05$; 3—$\bar{\rho }=0.10$; 4—$\bar{\rho }=0.15$; 5—$\bar{\rho }=0.2$.

**Figure 6 materials-15-00289-f006:**
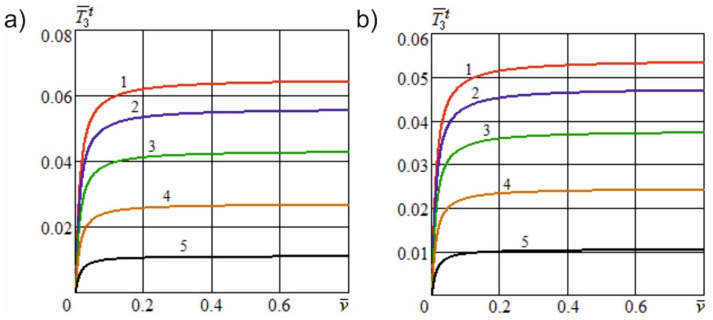
Dependence of the dimensionless useful resistance force on the dimensionless cutting speed at different angles of inclination of the cutting edge ($\bar{\rho }=0.2$; $e_{01}=5$; ${\bar{h}}_{u}=0.01$; ${\bar{l}}_{a}=0.4$): (**a**)—α=5°; (**b**)—α=10°; 1—$\gamma_{H}=0$; 2—$\gamma_{H}=25{}^{\circ}$; 3—$\gamma_{H}=40{}^{\circ}$; 4—$\gamma_{H}=55{}^{\circ}$; 5—$\gamma_{H}=70{}^{\circ}$.

**Figure 7 materials-15-00289-f007:**
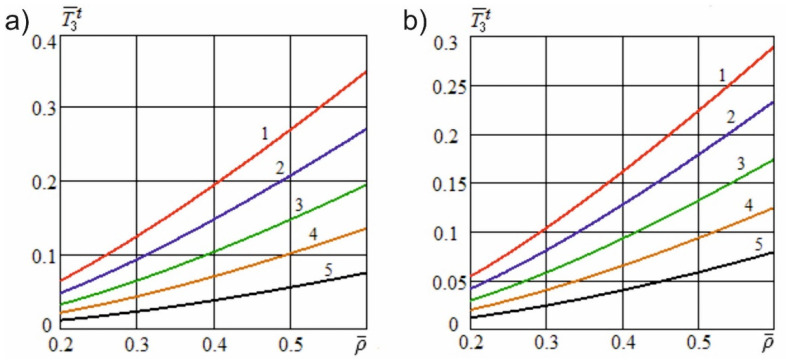
Dependence of the dimensionless useful resistance force on the dimensionless constructional sharpness of the knife at different angles of inclination of the cutting edge ($\bar{v}=5$; $e_{01}=5$; ${\bar{h}}_{u}=0.01$; ${\bar{l}}_{a}=0.4$): (**a**)—α=5°; (**b**)—α=10°; 1—$\gamma_{H}=0$; 2—$\gamma_{H}=35{}^{\circ}$; 3—$\gamma_{H}=50{}^{\circ}$; 4—$\gamma_{H}=60{}^{\circ}$; 5—$\gamma_{H}=70{}^{\circ}$.

**Figure 8 materials-15-00289-f008:**
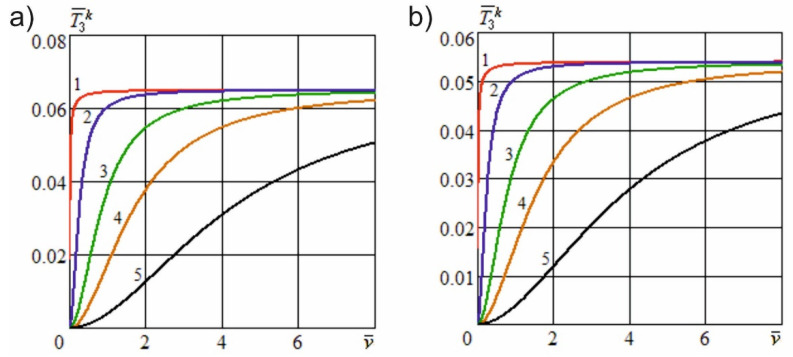
Dependence of the dimensionless useful resistance force on the dimensionless feed speed at various dimensionless sliding speed values ($\bar{\rho }=0.2$; $e_{01}=5$; ${\bar{h}}_{u}=0.01$; ${\bar{l}}_{a}=0.4$): (**a**)—α=5°; (**b**)—α=10°; 1—${\bar{v}}_{o}=0$; 2—${\bar{v}}_{o}=0.3$; 3—${\bar{v}}_{o}=1$; 4—${\bar{v}}_{o}=2$; 5—${\bar{v}}_{o}=5$.

**Figure 9 materials-15-00289-f009:**
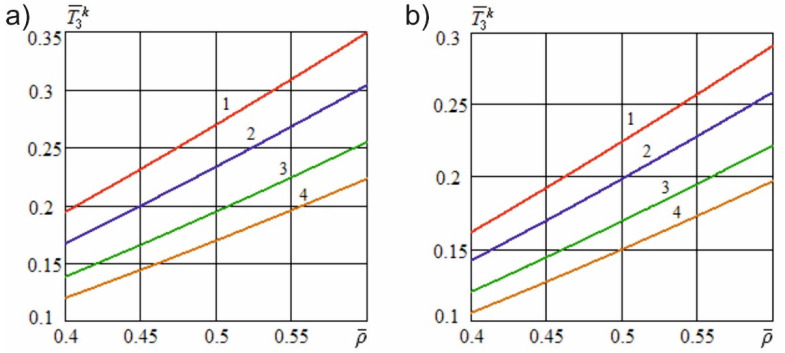
Dependence of the dimensionless useful resistance force on the dimensionless constructional sharpness of the knife at various dimensionless sliding speed values ($\bar{v}=5$; $e_{01}=5$; ${\bar{h}}_{u}=0.01$; ${\bar{l}}_{a}=0.4$): (**a**)—α=5°; (**b**)—α=10°; 1—${\bar{v}}_{o}=0$; 2—${\bar{v}}_{o}=2.5$; 3—${\bar{v}}_{o}=4$; 4—${\bar{v}}_{o}=5$.

**Figure 10 materials-15-00289-f010:**
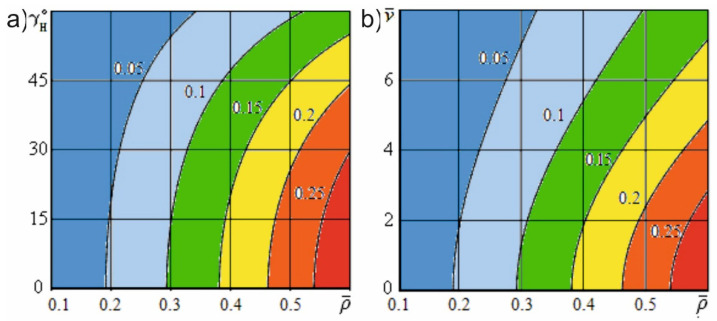
Dimensionless useful resistance force (α=10°; $e_{01}=5$; ${\bar{h}}_{u}=0.01$; ${\bar{l}}_{a}=0.4$): (**a**) dependence on the dimensionless constructional sharpness of the knife and the angle of inclination of the cutting edge ($\bar{v}=5$); (**b**) dependence on the dimensionless constructional sharpness of the knife and the sliding speed of the cutting edge (${\bar{v}}_{p}=5$).

## Data Availability

Data sharing is not applicable to this article.

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
