# Peer review of "Increasing the Efficiency of Food Material Cutting during Inclined and Shear Movements of Knife"

_materials, 2021, doi:10.3390/ma15010289_

Round 1

Reviewer 1 Report

Paper is acceptable after incorporate my minors comments

1- Give the equation number of the function F1 at line number 118, also explain this formula.
2-Same at line 134 and so on.
3- Explain the figure in detail
4-Incrase the resolution of the figure as they are blur, if these figures in color that will be very nice.  
5- More explanations of each figure are required. Short the caption of each figure.
6- Explain numeric results of the paper in conclusion section.
7- Improve overall the presentation of the paper.

Reviewer 2 Report

The work of this manuscript seems a litter interesting and has potential application. The authors investigated the dependence of the actual cutting angle at various conditions. It was proposed that the useful resistance force could be significantly reduced during the inclined and sliding cutting process. However, there are still some issues in the current manuscript. The manuscript will be probably considered for publication after addressing the following issues. 1) The section ‘Introduction’ is written too poorly and fails to effectively summarize the previous work related. 2) The roughness of the knife surface is also an important parameter, but it is not mentioned in this study. 3) In Page 11 line 267-270, the constructional knife sharpening angle is different (40 or 10), but the values of the actual cutting angle are the same. Why? 4) The section ‘Discussion’ is not in-depth, and most are the testing results simply listed. 5) The English writing needs great improvement so that the goals and results of this study are clear to the reader.
